December 15, 2020
**Temperature decadal trends, and their relation to diurnal variations in the lower**
**thermosphere, stratosphere, and mesosphere, based on measurements from SABER on**
**TIMED.**
**Frank T. Huang[1*], Hans G. Mayr[2*]**
[1]University of Maryland, Baltimore County, MD 21250, USA
[2]NASA Goddard Space Flight Center, Greenbelt, MD 20771, USA
*retired
**Abstract.** We have derived the behavior of decadal temperature trends over the 24 hours of local
time, based on zonal averages of SABER data, years 2012 to 2014, 20 to 100 km, within 48º of
the equator. Similar results have not been available previously. We find that the temperature
trends, based on zonal mean measurements at a fixed local time, can be different from those
based on measurements made at a different fixed local time. The trends can vary significantly in
local time, even from hour to hour. This agrees with some findings based on night-time lidar
measurements. This knowledge is relevant because the large majority of temperature
measurements, especially in the stratosphere, are made by instruments on sun-synchronous
operational satellites which measure at only one or two fixed local times, for the duration of their
missions. In these cases, the zonal mean trends derived from various satellite data are tied to the
specific local times at which each instrument samples the data, and the trends are then also
biased by the local time. Consequently, care is needed in comparing trends based on various
measurements with each other, unless the data are all measured at the same local time. A similar
caution is needed when comparing with models, since the zonal means from 3D models reflect
averages over both longitude and the 24 hours of local time. Consideration is also needed in
merging data from various sources to produce generic, continuous longer-term records. Diurnal
variations of temperature themselves, in the form of thermal tides, are well known, and are due
to absorption of solar radiation. We find that at least part of the reason that temperature trends
are different for different local times is that the amplitudes and phases of the tides themselves
follow trends over the same time span of the data. Much of past efforts have focused on the
temperature values with local time when merging data from various sources, and on the effect of
unintended satellite orbital drifts, which result in drifting local times at which the temperatures
are measured. However, the effect of local time on trends has not been well researched. We also
derive estimates of trends by simulating the drift of local time due to drifting orbits. Our
comparisons with results found by others (AMSU, lidar) are favorable and informative. They
may explain at least in part, the bridge between results based on daytime AMSU data and night
time lidar measurements. However, these examples do not a pattern make, and more
comparisons and study are needed.

**1.0 Introduction**
The understanding of decadal temperature trends in the middle and upper atmosphere is
interesting scientifically and important for practical reasons. Global temperature trends have
been researched for decades based on a variety of satellite and ground-based measurements.
However, relatively few studies have focused on the behavior of trends as a function of local
time. Past efforts have focused more on the local time variations of temperature themselves in
comparing or merging various data sets, and on accounting for drifts in local time of
measurements due to satellite orbital stability.
Diurnal variations of temperatures themselves, in the form of thermal tides, are well known,
and are a result of absorption of solar radiation (see Brasseur and Solomon [2005] and references
therein).
Understanding the behavior of trends with local time can be important because the large
majority of global temperature measurements, especially in the stratosphere, are made by sun-
synchronous satellites whose instruments measure temperature at only one or two fixed local
times, for the duration of their missions. In these cases, the zonal mean trends derived from
various satellite data are tied to, and biased by the specific local times at which each instrument
samples the data.
Care is then needed in comparing results of trends derived from various measurements which
sample data at different local times. It is also needed when merging data from various sources to
produce generic, continuous, longer-term records. In addition, the zonal means of 3D models are
averages of temperatures over both longitude and the 24 hours of local time, and comparisons
with trends based on data taken at fixed local times, or a subset of local times, can be
problematic (Austin et al., 2008).
In the following, based on data from the Sounding of the Atmosphere using Broadband
Emission Radiometry (SABER) instrument on the Thermosphere-Ionosphere-Mesosphere-
Energetics and Dynamics (TIMED) satellite, we derive the local-time dependence of decadal
temperature trends over the 24 hours of local time, from 2002 to 2014, from the stratosphere into
the lower thermosphere (20 to 100 km), within 48º of the equator.
Comparable results for temperature trends have not been available previously.
Our starting point here is based on results from our past studies, also based on SABER data.
Previously, we had estimated diurnal variations of the temperature (thermal tides) for each day,
expressed in the form of five Fourier series components (Huang et al., 2010a). We had also
derived zonal means of temperature that are averages over both longitude and local time for a
latitude circle (Huang et al., 2006, Huang et al., 2010a). These 'synoptic' zonal means are
important because they can then be compared directly with 3D models. Details are given in
Section 2.
Using these past results, we here derive the behavior of decadal temperature trends as a
function of local time.
We find that the temperature trends, based on zonal mean measurements at a fixed local time,
can be different from those based on measurements made at a different fixed local time. These
variations of trends can be significant in all regions of our study, and can vary significantly even
from hour to hour.
Our results suggest that part of the reason that temperature trends are different for different
local times is that the tidal amplitudes and phases of the tides also follow trends over the time
span of the data.
In the following, we compare with results of trends by others. Because trends vary with the
time span considered, comparisons should cover similar times, and the opportunities are limited.
Although the comparisons support our results of local time variations in trends, more
comparisons are needed.
Global stratospheric data are largely from the NOAA series of operational satellites and the
Earth Observing System of satellites. These are generally in sun-synchronous orbits, so that data
are sampled at only one or two local times, which are fixed for the duration of the missions. The
operational satellites are meant in part to monitor the atmosphere over the longer term, and have
been making measurements since the 1970s. Over the years, they are replaced as needed, in order
to maintain a continuous record of data. However, there have been issues of data continuity and
compatibility among the different satellites, related to data sampling, instrument calibration, and
operation. Also, over the years, the orbits of some satellites have drifted from their planned sun-
synchronous state, so that the local times at which the measurements are made have also drifted
over several hours or more.
There have been group and individual efforts to combine and merge the data from different
sources to obtain uniform, consistent, decades-long data bases for temperature (and others). Parts
of the issues are concerned with differences due to local times when merging data. For example,
Mears and Wentz [2016] have considered the sensitivity of temperature trends to "diurnal cycle
adjustment", and improved the consistencies of the different data sets caused by orbital drifts in
local time, based on cross information from other satellites, and on general circulation models.
Keckut et al., [2015] have also shown that considering atmospheric tides to account for
differences among measurements of successive operational polar orbiting satellites would
improve matters. Funatsu et al., [2008] have studied the differences among night time lidar data
and daytime sun-synchronous satellite data. Randel et al, [2016], McLandress (2015), Zou et
al.,[2014, 2016], among others, have also considered the issue of merged data from various
sources, with consideration for differences due to effects of local time
These merged long-term datasets have general advantages of providing for studies of trends
and responses to solar activity.  However, as noted earlier, if the various data sets do not
represent uniform sampling in local time, the merged data could be tagged by the biases in local
times.
**2.0 Previous results**
Because we make use of our previous results of temperature diurnal variations and trends, we
briefly review for the convenience of the reader. Our previous results on temperature trends were
based only on zonal means that are averages over both longitude and local time. See Huang et
al., [2014, 2010a].
The SABER instrument was launched in December 2001 on the TIMED satellite (Russell et
al., 1999). The data used here is version 2.0, level2A. The values are interpolated to 4-degree
latitude and 2.5 km altitude grids, and zonal averages are taken for analysis.
SABER temperature measurements have been analyzed with success by us and by others.
Zhang et al. [2006] and Mukhtarov et al. [2009] have derived temperature diurnal tides using
SABER data, and Nath and Sridharan [2014] have derived temperature trends using the same
SABER data, but without accounting for diurnal variations. We have derived variations with
periods from less than one day (diurnal variations) up to multiple years (semiannual oscillations
(SAO), quasi-biennial oscillations (QBO)), and one decade or more (responses to the solar
cycle). See Huang et al. [2010a, 2014, 2016a,b].
In a previous paper, Huang and Mayr [2019] had also analyzed the effects of local time on the
response of temperature (and ozone) to the solar cycle (~ eleven years).
**2.1 Diurnal variations**
Due to the orbital characteristics of TIMED, SABER measurements provide the potential to
estimate the variations of temperature as a function of the 24 hours of local time that data from
other satellites generally do not provide. The local times of the SABER measurements decrease
by about 12 min from day to day, and it takes 60 days to sample over the 24 hours of local time,
using both ascending and descending node data. Although this provides essential information
over the range of local times, over 60 days, variations can be due to both local time and other
variables, such as season. Diurnal and mean variations are embedded together in the data and
need to be unraveled from each other to obtain more accurate estimates of each.
Our algorithm is designed for this type of sampling in local time and provides estimates of
both diurnal and mean (e.g., annual, semiannual, seasonal oscillations) variations together in a
consistent manner. At a given latitude and altitude for zonal mean data over a period of a year,
the algorithm performs a least squares estimate of a two-dimensional Fourier series, where the
independent variables are local time and day-of-year, and variations as a function of local time
and day-of-year are generated.
The fundamental Fourier period in day-of-year is 365 days, and that for local time is 24 hours.
For subsequent months and years, the initial analysis serves as a sliding data window. To find
subsequent monthly values, this window is advanced by one month, and the algorithm is applied
again. Further details can be found in Huang et al.,[2010a].
Figure 1 shows an example of temperature diurnal amplitudes (left panel) and phases (right
panel) of the diurnal tide on altitude-latitude coordinates (20 to 100 km, 48ºS to 48ºN), for day
85 of 2009. Although not shown, higher components, such as the semidiurnal tides can also be
significant. Our derivation includes 5 Fourier components.

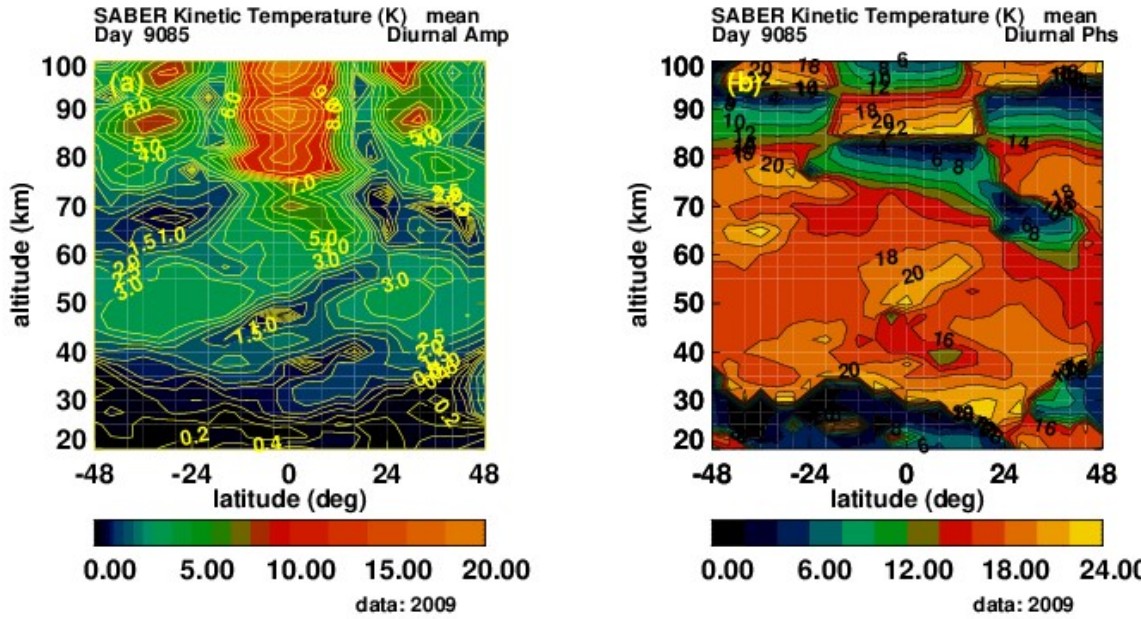

**Figure 1**. Temperature tides from 20 to 100 km, 48ºS to 48ºN, day 85, year 2009. Left panel (a): diurnal amplitudes
(K). Right (b): diurnal phases (hr maximum values).
**2.2. Mean variations.**
The zonal mean variations, which are averages over both longitude and local time, consistent
with 3D models, are obtained together with the diurnal variations.
Based on these zonal means, our earlier results of trends and decadal responses to solar activity
had been presented in Huang et al. [2014, 2016a, 2016b, 2019].

**3.0 Current analysis**
For the current analysis, we use the diurnal and mean variations together and generate zonal
means at any selected local time.
In the following, we generate
1)  Monthly zonal means that are averages over longitude, but at specific local times, to
correspond to measurements by sun-synchronous satellites and night-time lidar measurements.
2)  Monthly zonal means to simulate satellite orbital drifts, with local times that vary from month
to month.
3)  Monthly zonal means that are averages over longitude and the 24 hours of local time, as
previously done.
From 1), 2), and 3) we estimate temperature trends using Equation (1), in a similar manner as
previously done by others, and by us, using a multiple regression analysis that includes solar
activity, trends, seasonal, quasi biennial oscillations (QBO), and local time terms, on monthly
values. Specifically, the estimates are found from the equation

$$T(t) = a + b*t + c*S(t) + l*lst \ (t) + g*QBO(t) + d*F107(t) \qquad (1)$$

where t is time (months), a is a constant, b is the trend , $d$ the coefficient for solar activity (10.7
cm flux), c is the coefficient for the seasonal *(S(t))* variations,  $l$ the coefficient for local time *(lst)*
variations, and $g$ the coefficient for the QBO. As is often done, the seasonal and local time
variations are removed first, but we include them in Equation (1) for completeness. The F107
stands for the solar 10.7 cm flux, which is commonly used as a measure of solar activity, and the
values used here are monthly means provided by NOAA.
T stands for the various input temperature zonal means described in 1), 2), and 3), above.
The multiple regression is applied to the monthly zonal-mean values from June 2002 through
June 2014 from 48°S to 48°N latitude, and from 20 to 100 km.
The analysis of uncertainties is the same for this study as for the previous study of the mean
variations just described. Here the zonal means are generated at specific local times. Details and
results of the statistical analysis are given in Huang et al.,[2014, 2106a].
**4.0 Current results: temperature trends as a function of local time**
Before presenting our overall trend results as a function of local time, we first compare some
specific results with those by others. The merged data sets noted earlier do not represent uniform
averages over the range of local times nor do they represent specific fixed local times. In
addition, they span a longer time interval than the SABER data, and we will not use them for
comparisons. Because trends can significantly depend on the particular time period, comparisons
are limited to the time span ~ 2002 to 2014.
Figure 2(a) (left panel) shows examples of our estimates of monthly SABER values of
temperature (K) from mid 2002 to mid 2014, without the diurnal and seasonal variations. The
black line shows zonal mean values that are averages over both longitude and local time at 40
km and 16º latitude, with a trend of ~ -0.6 K/decade, found from a linear fit. The red line shows
monthly values of zonal means at a fixed 12 hrs local time, with a trend of ~ - 0.91 K/decade.
The blue line represents monthly values of zonal means at a fixed local time of 18 hrs, with a
trend of ~ + 0.94 K/decade.  Figure 2(b) (right panel) shows the temperature tidal diurnal
amplitude (black line, left hand scale) and the diurnal phase (red line, hour of maximum value,
right hand scale).
The trends of the diurnal amplitudes and phases themselves contribute to the different
temperature trends at different local times. Although not shown, we note that semidiurnal tides
are not negligible. Additional plots corresponding to Figure 2(b) are given in the Appendix.

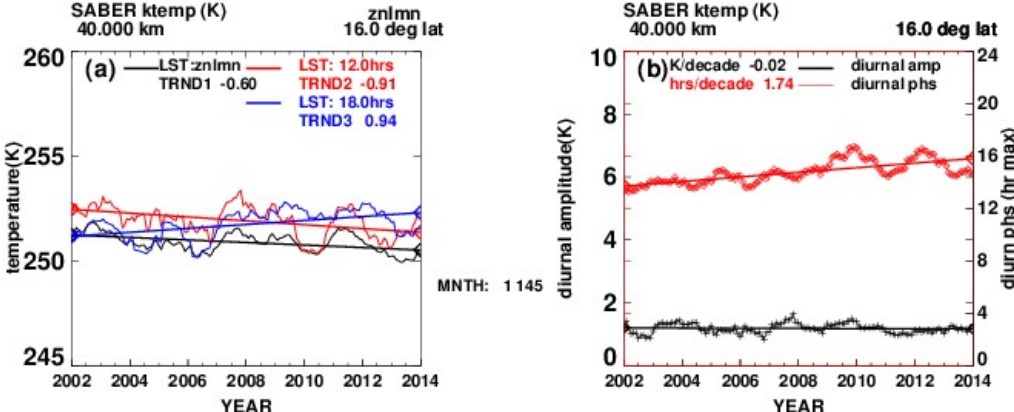

**Figure 2.** Left panel (a): Monthly SABER temperature (K) from 2002 to 2014, 40 km, 16ºN latitude. Black line:
zonal mean values (averages over longitude and local time); red line: zonal mean at 12 hrs local time; blue line:
zonal mean at 18 hrs local time. Right panel (b): left axis scale: black line: tidal diurnal amplitude (K); red line, right
axis scale: diurnal phase (hr of maximum value).
**4.1 Stratosphere.**
For the stratosphere, we compare with trends given by Funatsu et al.,[2016], based on data
from the Advanced Microwave Sounding Unit (AMSU) on the NASA Aqua satellite and from
night-time ground-based lidar measurements. The results of Funatsu et al.,[2016] are suitable for
comparison because the time span of the data are similar to ours (2002 to 2014), and AMSU
samples data near specific local times, namely, 13:30 and 1:30 local times.
Following Funatsu et al.,[2016], the AMSU is a cross-scanning microwave-based sounder and the
channels 9–14 sample with weighting functions peaking at approximately 18, 20, 25, 30, 35, and 40
km. The horizontal resolution at the near-nadir field of view is approximately 48 km, and the vertical
half width of the weighting functions is about 10 km.
Although the lidar measurements presented by Funatsu et al.,[2016] also cover a similar time
span (2002-2013), they are made only during night time from the Observatoire de Haute
Provence (OHP, 43.91ºN, 5.71ºE) and the Mauna Loa Observatory (MLO, 19.51ºN, 155.61ºW).
Figure 3 shows our results of temperature trends (K/decade) based on SABER data (2002 to
2014) and those from Funatsu et al.,[2016], based on AMSU and lidar measurements. For
AMSU, Fanatsu et al., [2016] provide trends as a function of channel numbers for the low and mid
latitude composite trends, so following McLandress et al.,[ 2015], we use the altitudes of the
weighting function peaks, namely 20, 25, 30, 35, and 40 km, for comparison. They do provide
altitudes in km for comparison with lidar. We note that where the values and altitudes are given by
others such as Funatsu et al., [2016], we have transferred them manually to our figures, as needed.
In the top left panel (a) of Figure 3, the black line plots trend results based on SABER zonal
means found by averages over both local time and longitude. The blue diamonds and squares are
from Funatsu et al.,[2016], based on AMSU data, presumably averages taken near 1:30 and
13:30 hours. The blue diamonds denote zonal mean trends for mid latitudes (30º to 60ºN), and
the blue squares represent trends at 44ºN to correspond to OHP. The blue squares are available
only from 30 to 40 km (~ 30, 32.0, 36.2, 40.0 km), but match our results (black line) at 44ºN
extremely well. The blue diamonds (from Funatsu AMSU, an amalgam to represent mid latitude)
match our results almost exactly from 20 to 30 km, but are larger from 30 to 40km. This could
simply be that the blue diamonds represent mid latitudes (30º to 60ºN) while the blue squares
and our black line represents trends at 44ºN specifically. The magenta asterisks, also provided by
Funatsu et al.,[2016], based on night-time lidar measurements at 44ºN, are significantly more
negative from 30 to 40 km than our results and those of the Funatsu et al.,[2016] AMSU. The top
right panel (b) of Figure 3 shows our night time results from SABER at 21, 22, 23 hrs. It can be
seen that our night time results agree better with the night-time lidar trends (magenta asterisks) in
the left panel Figure 3(a). We do not know the details of the night-time hours of the lidar data.
The bottom row left panel (c) of Figure 3 shows our daytime trends at 9, 10, 11 hrs, and agree
less well with the lidar trends.
The average of all our night and daytime trends gives the zonal mean average shown by the
black line. The bottom right panel (d) compares our results at 1, 2, 13, and 14 hrs with the
AMSU results. They are near the local times of the AMSU data (presumably 1:30 and 13:30 hrs).
It can be seen that the averages over the 4 local times compare favorably with those of Funatsu et
al., [2016], based on AMSU data. It is not clear if Funatsu et al.,[2016] differentiated night from
day measurements.
We believe that, by taking into account trends with local time, our results compare favorably
with both the Funatsu et al., [2016] AMSU trends and their results based on night time lidar data.

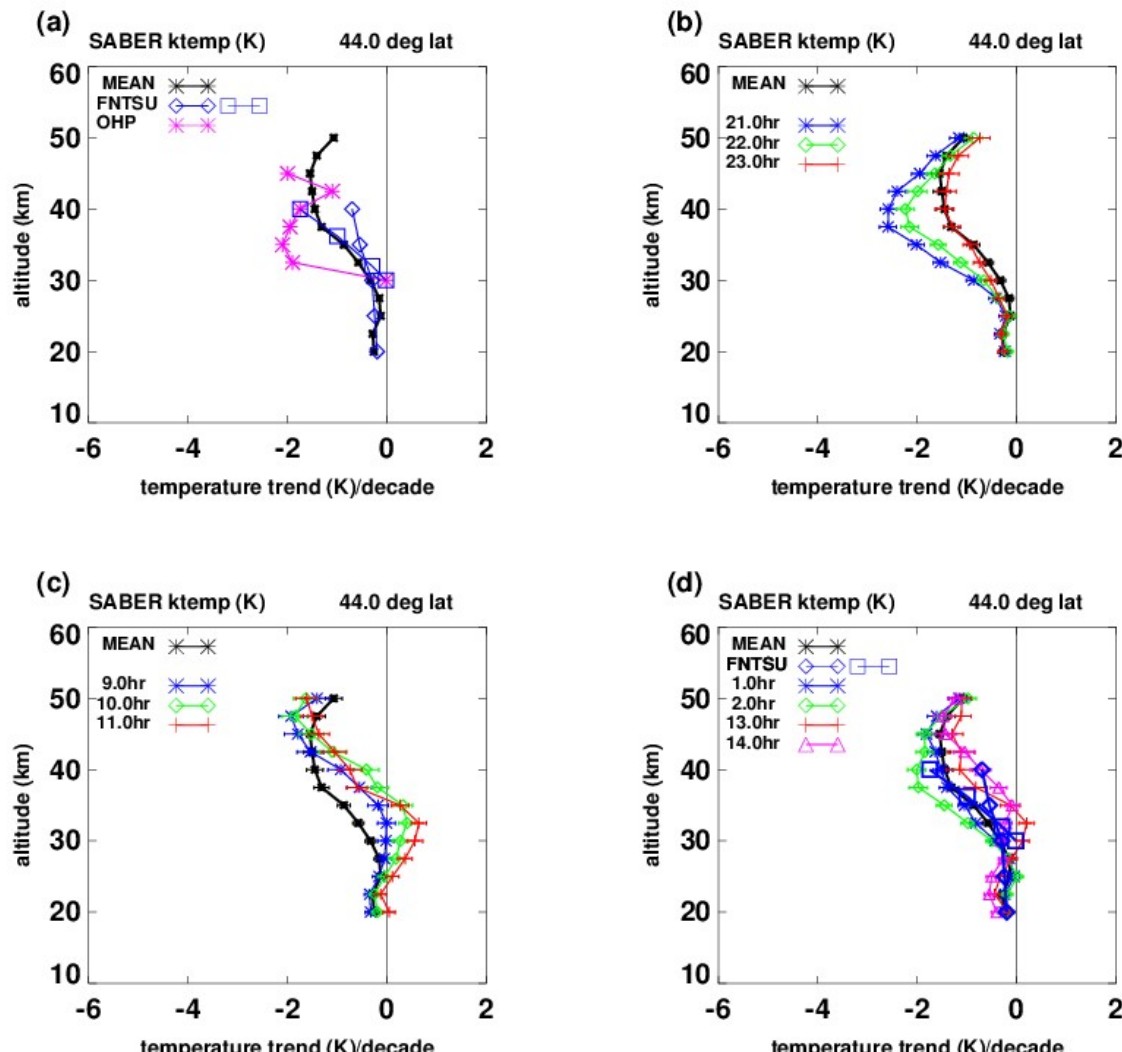

**Figure 3.** Temperature trends (K/decade, 2002-2014) vs altitude. Top left (a): Black asterisks:based on  SABER
zonal means  (over longitude and local time) at 44ºN; blue diamonds: Funitsu Aqua trends for mid latitudes (30º-
60ºN); blue squares: Funatsu Aqua trends at 44ºN; magenta asterisks: based on night-time lidar measurements at
OHP (44ºN).  Top right (b): Black asterisks: same as (a), blue, green, red: our estimates at 21, 22, 23 hrs local time,
based on SABER data. Bottom left (c): Black asterisks: same as (a), blue, green, red: our estimates at 9, 10, 11 hrs
local time; bottom right (d): Black asterisks, same as (a);  blue diamonds and squares: as in panel (a), Funatsu
AMSU, blue asterisks, green diamonds, red plusses, magenta triangles: SABER trends at 1, 2, 13, 14 hours.

The left panel of Figure 4 corresponds to that of Figure 3, but for 20ºN to compare with
results of Funatsu et al., [2016] based on AMSU low-latitude and night-time lidar results at the
Hawaiian Mauna Loa Observatory (MLO, 19.51ºN). As in Figure 3 for OHP, the lidar results
show a diversion to more negative trends near 30-35 km. Here, our results, as represented by
trends based on zonal means that are averages over local time also show a decrease, although not
as pronounced, near 30-35 km. As in Figure 3, both the blue diamonds and blue squares are from
Funatsu et al., [2016] based on AMSU data, but for low latitudes (0 to 30ºN), and 20ºN latitude,
respectively. They are smoother than our results between 25 and 40 km and do not show the
notch near 30 km that we and the lidar-based trends show. This could be due to the differences in
altitude resolution between AMSU and lidar and SABER data.
As can be seen in the right panel (b) of Figure 4, the decrease on our trends near 30 km is due
in large part to the behavior at 21 and 22 hours (green diamonds, red plusses).
Figures 3 and 4 show that by taking into account the different trends with local time, our
results compare more favorably with those of the Funatsu et al., [2016], based on AMSU  and
lidar data.  Figures 3 and 4 also show that trends can change significantly with local time, even
from hour to hour.
However, our comparisons do not a pattern make, and more comparisons are of course
needed.
We note that the results of Khaykin et al.,[2017] based on analysis of GPS Radio Occultation
(GRO) measurements are in excellent agreement with AMSU (based on a slightly longer period
(2002-2016).  Khaykin et al.,[2017] state that," after down sampling of GRO profiles according
to the AMSU weighting functions, the spatially and seasonally resolved trends from the two data
sets are in almost exact agreement with trends based on AMSU data."

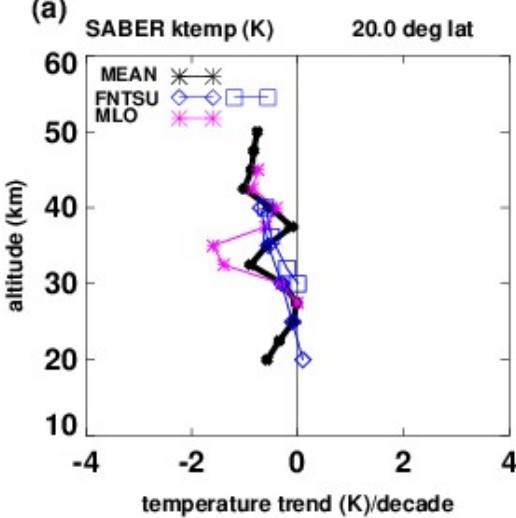 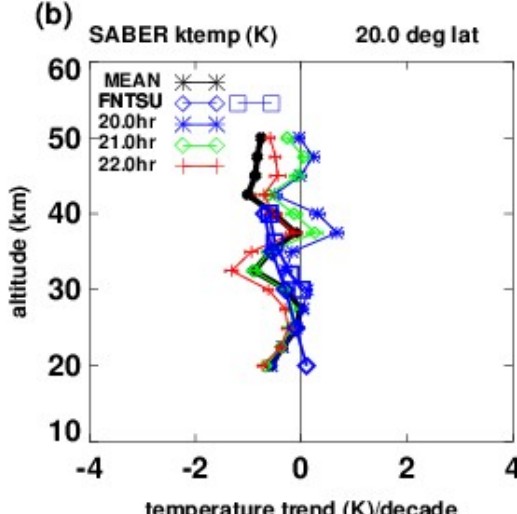

**Figure 4.** Temperature  trends (K/decade)  vs altitude. Left (a): Black asterisks:trends based on  SABER zonal
means (over longitude and local time) at 20ºN; blue diamonds: Funitsu et al.,[2016] Aqua; data at 13.5 and 1.5 hrs,
low latitudes (0º to 30ºN); blue squares: Funitsu Aqua at 20ºN; magenta asterisks: lidar measurements at  Mauna
Loa Observatory (MLO, 19.51ºN), Right (b): Black asterisks: same as (a), blue, green, red: estimates at 20, 21,
22 hrs local time, based on SABER data.
**4.2 Lower Thermosphere**
In Figure 5, we compare our results (K/decade) with the lidar night-time measurements of She
et al.,[2019], at Fort Collins, CO. (41ºN, 105ºW)/Logan Utah (42ºN, 112ºW), from  2002-2014.
They actually made nocturnal temperature observations between 1990 and 2017, but divided
their analysis into various time periods, and smaller time intervals within the night time hours.
This provides valuable information regarding trends and local time. In the left panel (a) of Figure
5, the magenta squares denote the mean night time trends derived by She et al.,[2019]. The black
line represents our trend results based on zonal means (averages over longitude and local time),
while the blue asterisks, green diamonds, and red pluses show our zonal mean trends at 19, 20,
and 21 hours, respectively. In contrast, the right panel (b) of Figure 5 shows corresponding
results based on Saber data in the day time at 15, 16, 17 hours local time. We have not included
more local times due in part that the plots become busy, and some lines reach maximum and
minima at different altitudes. Overall, the averages of day time and night trends result in the
black line.
It can be seen in Figure 5 that, as in Figures 3 and 4, changes in trends over as little as an
hour of local time can be significant. These results show that there are systematic differences in
derived trends at different local times. This agrees with those of She et al., [2019], who have also
derived trends averaged over 2 hrs at midnight, and they are significantly different from those
found from the all-night mean measurements. She et al., [2019] provide midnight results only for
a much larger time span (March 1990 to December 2017), so we do not compare.
Considering that the lidar data are not zonal means, and the details of the night-time sampling
are probably different from ours, we believe that our results generally support those of She et al.,
[2019].

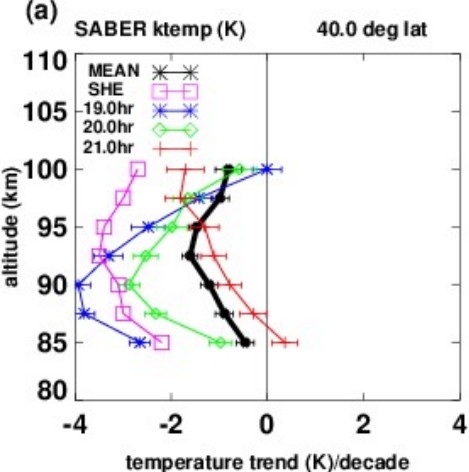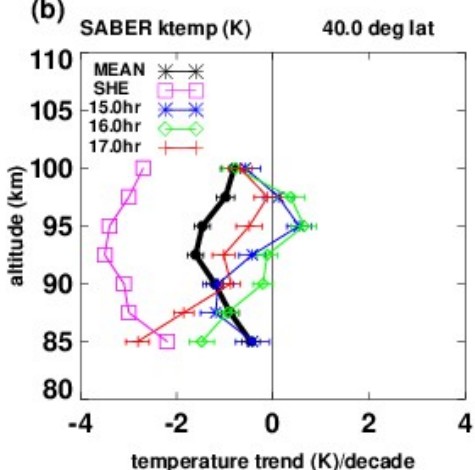

**Figure 5.** Temperature  trends (K/decade) vs altitude, at 40°N latitude. Left (a): Black asterisks: trends based on
SABER zonal mean (over longitude and local time); blue asterisks, green diamonds, red pluses: trends based on
SABER zonal means at 19, 20, 21 hrs local time. Magenta squares: trends based on night-time lidar measurements
by She et al.,[2019]; Right (b): as in (a) but for SABER results at 15, 16, 17 hrs local time.
**4.3 General results and orbital drift.**
Figure 6 shows more generally our derived trends (K/decade) at 20°N (left panel) and 44°N
(right panel), from 20 to 100 km, at different local times. The blue asterisks, green diamonds, red
pluses, and magenta triangles represent 0, 6, 12, and 18 hrs, respectively. The salient features
are that the trends can vary significantly as a function of local time, even from hour to hour.
Because temperature trends can depend on the time period of the data or models, so may tidal
trends. So we should not assume that the local time behavior of trends for different time periods
will be necessarily consistent with each other.
A broader and more detailed understanding would entail numerical studies, such as models
which include studies of trends relating to local times. Then trends in thermal tides could be
constrained to be zero to test effects on trends of the temperature.

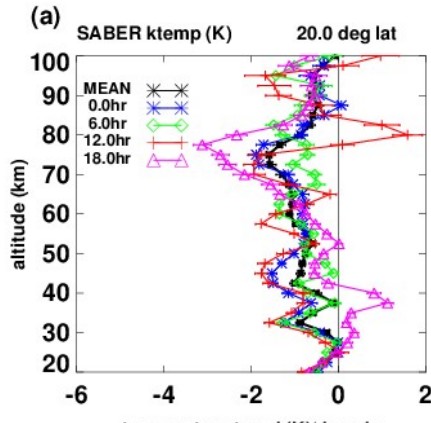 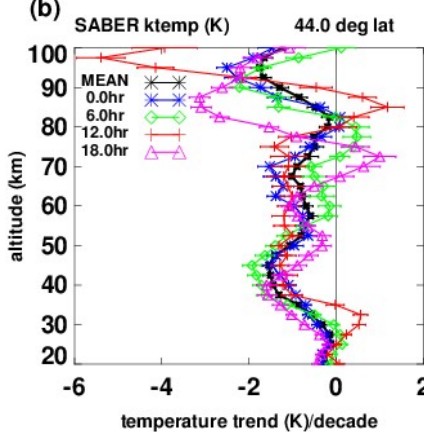

**Figure 6.** Temperature  trends (K/decade) vs altitude from 20 to 100 km at 20ºN (left panel) and 44ºN (right panel).
Black: trends based on SABER zonal means over longitude and local time; blue: based on zonal means at 0 hr;
green: 6 hrs, red: 12hrs, magenta: 18 hrs local time.
As noted earlier, over years, the orbits of some operational satellites have drifted from their
intended sun-synchronous state, so that the local times at which measurements are made have also
drifted, by several hours. As a simple simulated example, Figure 7 shows our results for temperature
trends (K/decade) versus altitude, at the Equator (left panel) and at 36ºN, from 20 to 60 km. The red
squares denote trends where local times increased linearly from 12 to 18 hrs from 2002 to 2014, to
simulate orbital drift. Black asterisks denote trends based on SABER data.
To our knowledge, there have been no previous similar results on this subject, and Figure 7 is
meant to provide only an indication of what may result when local times at which measurements are
made are not controlled.  Specifics would depend on the orbital drift of the particular satellite
(Funatsu et al., 2016) and on the particular study.

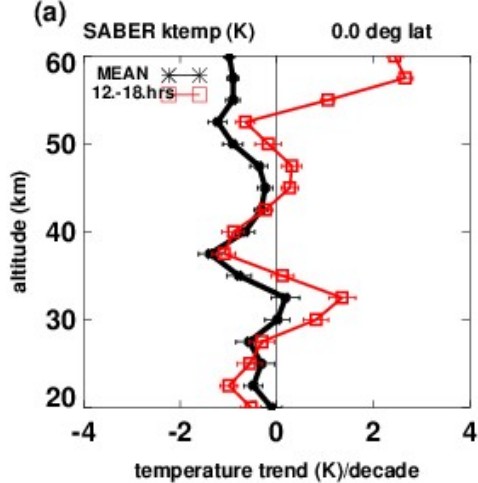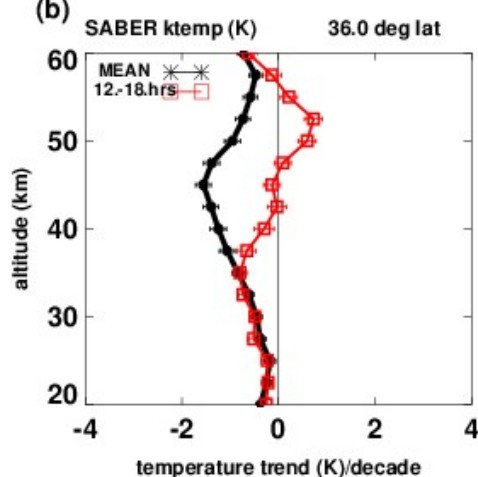

**Figure 7.** Temperature trends (K/decade) vs altitude at the equator (left panel) and 36ºN latitude (right panel). Black lines: trends based on SABER data (averaged over longitude and local time); red squares: estimated trends for cases where local times of measurements increase linearly from 12 to 18 hrs from 2002 to 2014.

## 5.0 Summary and conclusion

Using SABER data, we have investigated the local time variations of temperature trends (K/decade) from 2002 to 2014, 20 to 100 km, and 48ºS to 48ºN latitude. SABER provides global temperature measurements over the 24 hrs of local time, and from 20 to 100 km in altitude, that are not available from other satellites and sources.

From our past studies based on SABER data, we had estimated diurnal variations of the temperature (thermal tides) for each day, expressed in the form of five Fourier series components (Huang et al., 2010a).

We had also derived zonal means of temperature that are averages over both longitude and local time for a latitude circle (Huang et al., 2006, 2010a). These 'synoptic' zonal means are important because they can then be compared directly with 3D models (Austin et. al., 2008).

As explained earlier, zonal means from sun-synchronous satellites are tied to one or two local times. To our knowledge, comparable zonal means of temperature that are averages over longitude and the 24 hours of local time are just not available elsewhere.

In this current study, we have combined the past results to estimate the zonal mean trends corresponding to specific local times.

These results at local times have not been available previously. They show that the values of temperature decadal trends for a fixed local time are different from trends at another fixed local time. We find that the amplitudes and phases of the tides themselves also display decadal trends and are then likely contributors to the local time variations of temperature trends.

Our results of trend variations with local time are supported by comparisons with corresponding nighttime lidar measurements in the stratosphere and lower thermosphere. They are also supported by comparisons with corresponding satellite measurements made at specific local times in the stratosphere.

The dependence of trends on local time is significant throughout the region of analysis, and can
be significant even from hour to hour, as can be seen in Figures 3, 4, 5, and 6.
In the lower thermosphere, this agrees with corresponding trend results by She et al.,[2019],
based on lidar night-time measurements. She et al., [2019] found that trends based on a two-hour
average near midnight show systematic differences from the average over other hours. Our
comparisons with the overnight results of She et al., [2019] are seen in Figure 5, where our
trends at 19, 20, and 21 hours compare favorably, while our day time trends at 15, 16, and 17
hours compare less favorably.
In the stratosphere, our comparison with trends found by Funatsu et al.,[2016], based on lidar
and AMSU measurements, are even better, as seen in Figures 3 and 4.  At 44ºN (AMSU and
OHP lidar), Funatsu et al., [2016] provide AMSU trend results only from 30 to 40 km, but they
match our results almost exactly. Their results from 20 to 40 km, representing mid latitudes (30º
to 60ºN) also match our results almost exactly from 20 to 30 km, but are larger from 30 to 40km.
Between ~ 30 to 40 km, the night-time lidar trends are significantly smaller (more negative) than
both our and that of Funatsu et al.,[2016]. However, when the comparison is between night time
lidar and our night-time results (21, 22, and 23 hours, see Figures 3a, 3b), the agreements are
better.  At 20ºN (AMSU and MLO lidar), similar comments apply.
These examples all suggest that at least some of the differences between night time lidar
trends and those based on other measurements that are not made at night, can be explained at
least partly, through variations of trends with local time.
However, we emphasize that our three examples of course do not a pattern make, and more
direct comparisons are needed. Our current comparisons are limited because the various results
should be based on the similar time spans, and also not based on merged data from various
sources, as the identity in local time would not be clear for merged data.  Although there have
been previous studies related to variations with local time, they focused on mitigating differences
when merging data from different sources, and on accounting for temperature variations with
local time due to orbital drifts.
Because our results show that the data sets representing measurements at different fixed local
times can result in varying trends, merging those data can result in trends that cannot be tied to
specific local times, or to averages over the 24 hours of local time, as in 3D models, and can
result in biases.
**Appendix**
We present additional figures, corresponding to Figure 2(b), of temperature diurnal amplitudes
and phases over more altitudes (20, 40, 60, 80, 90 km) and latitudes (0º, 40º).
The left panels (a) of each figure show temperature tidal diurnal amplitudes and phases at
various altitudes and the Equator, while the right panels (b) correspond to the left panels but at
40ºN latitude.  In each panel, the left axis scale and black line denote tidal diurnal amplitudes
(K), while the right axis scale and red line show the diurnal phases (hr of maximum value).
The displayed trend values are obtained from a simple least squares straight line fit.   The larger
variations generally reflect modulation of the tides by the quasi biennial oscillation (QBO). This
has been discussed in models (Mayr and Mengel, 2005) and other SABER data (Forbes et al.,
2008).
Although the figures may give additional insight to the nature of the trends, there are caveats to
be considered. We note that the semidiurnal amplitudes and phases can also be significant. We
have derived a total of five Fourier components, and our numerical results reflect all 5 Fourier
terms.
Because both amplitudes and phases exhibit trends, they need to be considered in parallel, in
tandem, and this is difficult to discern, qualitatively. In addition, because the trends are generally
small, it would be difficult to arrive at conclusions.

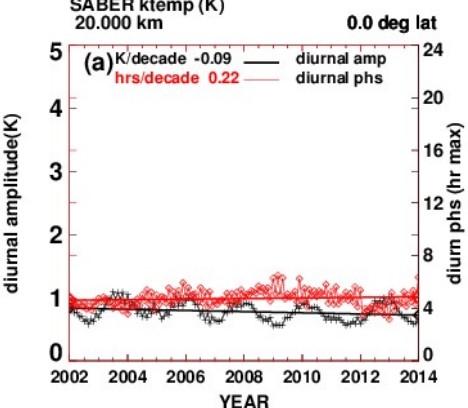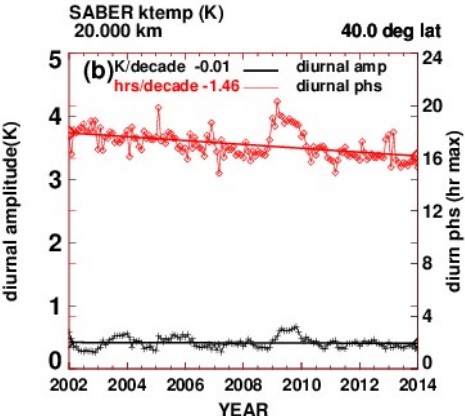

**Figure A1.** Left panel (a): Temperature tidal diurnal amplitudes and phases at 20 km and equator; left axis scale:
black line: tidal diurnal amplitude (K); right axis scale: red line: diurnal phase (hr of maximum value). Right panel
(b): as in left panel but at 40ºN latitude.

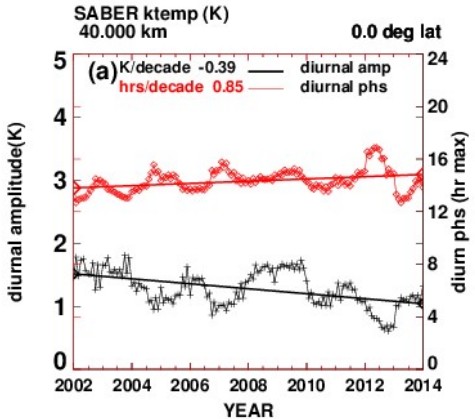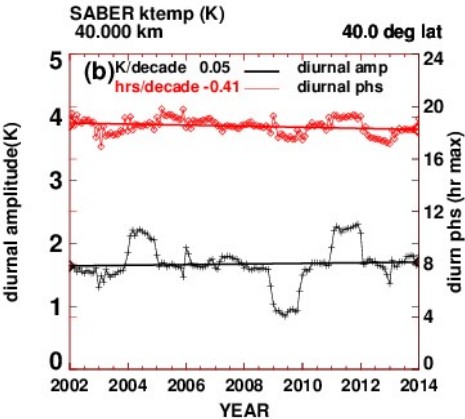

**Figure A2.** Left panel (a): Temperature tidal diurnal amplitudes and phases at 40 km and equator; left axis scale:
black line: tidal diurnal amplitude (K); right axis scale: red line: diurnal phase (hr of maximum value). Right panel
(b): as in left panel but at 40ºN latitude.

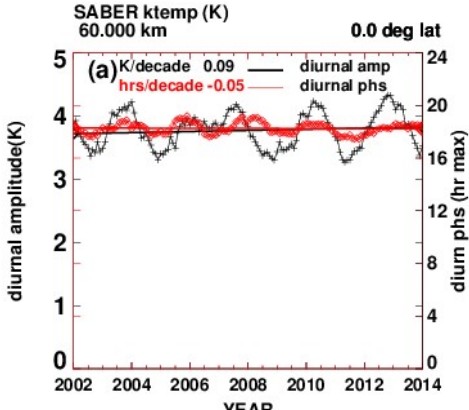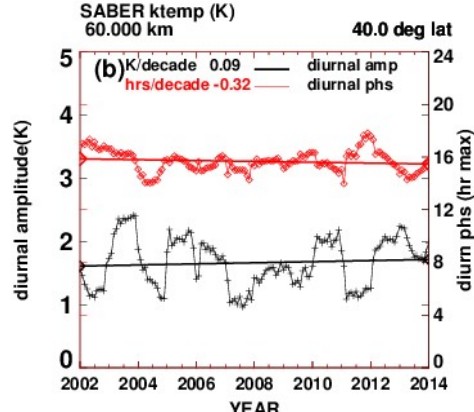

**Figure A3.** Left panel (a): Temperature tidal diurnal amplitudes and phases at 60 km and equator; left axis scale:
black line: tidal diurnal amplitude (K); right axis scale: red line: diurnal phase (hr of maximum value). Right panel
(b): as in left panel but at 40ºN latitude.

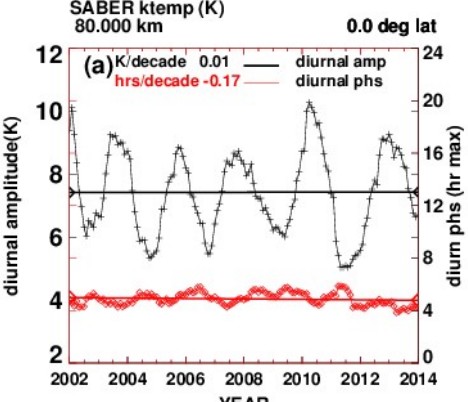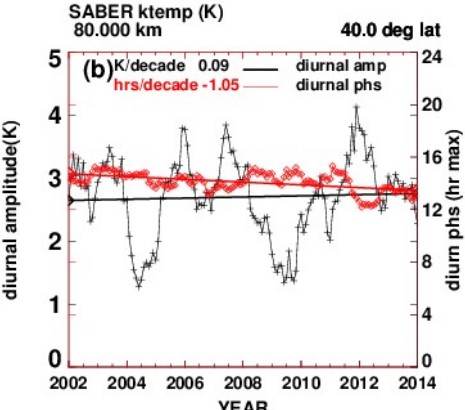

**Figure A4.** Left panel (a): Temperature tidal diurnal amplitudes and phases at 80 km and equator; left axis scale:
black line: tidal diurnal amplitude (K); right axis scale: red line: diurnal phase (hr of maximum value). Right panel
(b): as in left panel but at 40ºN latitude.

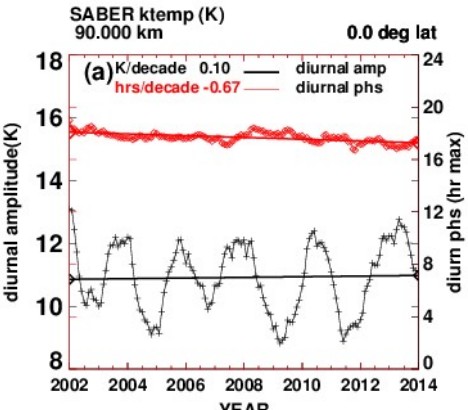 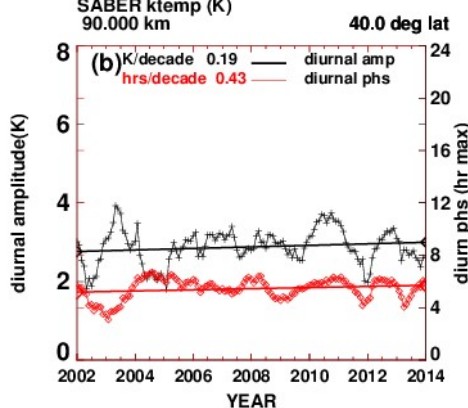

**Figure A5.** Left panel (a): Temperature tidal diurnal amplitudes and phases at 90 km and equator; left axis scale:
black line: tidal diurnal amplitude (K); right axis scale: red line: diurnal phase (hr of maximum value). Right panel
(b): as in left panel but at 40ºN latitude.

## Data availability
The SABER data are freely available from the SABER project at http://saber.gats-inc.com/.

**Acknowledgements.** We thank the editor G. Stober and two anonymous referees, whose
comments helped to improve the manuscript.

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
