# Peer review of "December 15, 2020"

_Annales Geophysicae, 2020_

## Referee Comment (RC1) · Anonymous Referee #1 · 6 Nov 2020

This manuscript presents a study on the relationship between diurnal variations in the temperature and its decadal trends from the stratosphere to the lower mesosphere, based on SABER measurements on the TIMED satellite. This study is useful because it is important to understand the evolution of temperature in the middle and upper atmosphere in relation to global warming at the surface, but the analysis of satellite observations can be biased by their non-uniform local solar time. Most satellites are sun-synchronous, always measuring at two fixed local times for a given latitude. In

addition, some of these satellites are affected by local time drifts during their lifetime. SABER is the only satellite providing temperature data at all local times in the mesosphere and the lower thermosphere, making it possible to separate the contribution of diurnal tides in temperature evolution. This study indicates that the estimation of temperature trends is not only biased by local time sampling but also by trends in the amplitude and phase of thermal tides. This is a new result that deserves to be published in Annales Geophysicae. However, I recommend to improve the content and the organization of the manuscript to make it more convincing as explained below.

The separation between diurnal variations and the long term evolution in temperature measurements by SABER is made using a least squares fit of a two dimensional Fourier series. This method is explained in details in Huang et al. (2010) and only very shortly described in the present paper. Such a detailed description is not needed in the present paper. However it would be useful for the readers to give a more complete synthesis of the method. In particular some parameters have to be fixed in the analysis as for instance the time length of the 2-D Fourier series, fixed to one year in Huang et al. (2010). Is it the same in the present study? It is important to know these parameters to understand the signification of decadal trends in mean temperature and in thermal tides.

The analysis provides the characteristics of the tides and their evolution over time. There is a detailed discussion of comparisons of temperature trends inferred from SABER data with published results from terrestrial lidars and AMSU satellites for several fixed local hours. These comparisons are very interesting and show a good general agreement, but results on the decadal evolution of tidal parameters (amplitude and phase) are not presented, except for the example given in Figure 2. I do not know any published results on the evolution of tidal parameters in the middle atmosphere and this would be a very valuable result.

Concerning the organisation of the paper, it is rather unusual to give some conclusions in the introduction section as it is done in lines 81-89. This part should be moved to the

summary and conclusion section.

---

## Referee Comment (RC2) · Anonymous Referee #2 · 1 Dec 2020

The authors have used temperatures obtained from SABER data to extend study about the local time variations of temperature trends from 2002 to 2014, 20 to 100 km, and 48°S to 48°N latitude. The trends found have been compared with those published and after some discussion the authors found that temperature decadal trends for a fixed local time were different from trends at another fixed local time. In addition, the authors also found that the amplitudes and phases of the tides also revealed decadal trends, and they inferred that thermal tides likely count to contribute to the local time

variations of temperature trends.

Based on my evaluation, this is an important scientific contribution which can help to clarify differences and achieve more consistent trend results. However, there are some concerns that need to be addressed.

1)The introduction can be improved in order to clarify the findings in previous works and the difference from present investigation. The authors have discussed this issue, however the way it is written has become confused.

2)One of the main results found by the authors concerns the contribution of thermal tides to the local time variations of temperature trends, since their amplitudes and phases also display decadal trends. In this sense, Figure 2b should be further explored in order to show readers how the variation in thermal tides contributes to the local time variations of temperature trends. The same can be considered in sections 4.1 (Stratosphere) and 4.2 (Lower Thermosphere), where the relationship between the thermal tides and the trends should be emphasized.

3)The diurnal tide on temperature in stratosphere ($\sim$ 40-50 km) has its strongest amplitudes at latitudes 40°N, S, while in the MLT region ($\sim$ 85-95 km) they are strongest around the equator, where the phases are best resolved. Therefore, I would like to suggest adding the analysis of trends in the amplitudes and phases of the diurnal tide for these regions, which may provide enrichment in the discussion of results.

4)The theme of the section 4.3 (Orbital drift and generic) needs to be improved and explored. It could be moved to section 2 without prejudice to the manuscript.

5)Discussion should be made more rigorous. The basis for the statements needs to expand further.

Technical revision

Please check the citations through the text considering the rules required by the journal.

Page 2, line 82: change "Advanced Microwave Sounder..." by "Advanced Microwave Sounder Unit"

Page 7, Figure 3b,c,d: legend overlap with plots

Page 11, Figure 7: legend overlap with plots

---

## Author Comment (AC1) · 14 Dec 2020

December 14, 2020

Authors' response to referee#1 concerning manuscript titled "Temperature decadal trends, and their relation to diurnal variations inthe lower thermosphere, stratosphere, and mesosphere, based on measurements from SABER on TIMED"

[Figure]

by Frank T. Huang and Hans G.Mayr

ANGEOD Interactivecomment Discussion paper Ann. Geophys. Discuss.,https://doi.org/10.5194/angeo-2020-63-RC1, 2020© Author(s) 2020. This work is distributed underthe Creative Commons Attribution 4.0 License.

Interactive comment on "Temperature decadaltrends, and their relation to diurnal variations inthe lower thermosphere, stratosphere, and mesosphere, based on measurements fromSABER on TIMED"

Anonymous Referee #1

1) Refereee #1: This manuscript presents a study on the relationship between diurnal variations in the temperature and its decadal trends from the stratosphere to the lower mesosphere, based on SABER measurements on the TIMED satellite. This study is useful because it is important to understand the evolution of temperature in the middle and upper atmosphere in relation to global warming at the surface, but the analysis of satellite observations can be biased by their non-uniform local solar time. Most satellites are sun-synchronous, always measuring at two fixed local times for a given latitude. In addition, some of these satellites are affected by local time drifts during their lifetime. SABER is the only satellite providing temperature data at all local times in the mesosphere and the lower thermosphere, making it possible to separate the contribution of diurnal tides in temperature evolution. This study indicates that the estimation of temperature trends is not only biased by local time sampling but also by trends in the amplitude and phase of thermal tides. This is a new result that deserves to be published in Annales Geophysicae. However, I recommend to improve the content and the organization of the manuscript to make it more convincing as explained below.

2) Referee #1: The separation between diurnal variations and the long term evolution in temperature measurements by SABER is made using a least squares fit of a two dimensional Fourier series. This method is explained in details in Huang et al. (2010) and only very shortly described in the present paper. Such a detailed description is not

needed in the present paper. However it would be useful for the readers to give a more complete synthesis of the method. In particular some parameters have to be fixed in the analysisas for instance the time length of the 2-D Fourier series, fixed to one year in Huang etal. (2010). Is it the same in the present study? It is important to know these parameters to understand the signification of decadal trends in mean temperature and in thermal tides.

Authors' response to 2):

We have added the following to Section 2.1 of the manuscript, as follows:

"Due to the orbital characteristics of TIMED, SABER measurements provide the potential to estimate the variations of temperature as a function of the 24 hours of local time that data from other satellites generally do not provide. The local times of the SABER measurements decrease by about 12 min from day to day, and it takes 60 days to sample over the 24 hours of local time, using both ascending and descending node data. Although this provides essential information over the range of local times, over 60 days, variations can be due to both local time and other variables, such as season. Diurnal and mean variations are embedded together in the data and need to be unraveled from each other to obtain more accurate estimates of each. Our algorithm is designed for this type of sampling in local time and provides estimates of both diurnal and mean (e.g., annual, semiannual, seasonal oscillations) variations together in a consistent manner. At a given latitude and altitude for zonal mean data over a period of a year, the algorithm performs a least squares estimate of a two‐dimensional Fourier series, where the independent variables are local solar time and day-of-year, and variations as a function of local time and day-of-year are generated. The fundamental Fourier period in day-of-year is 365 days, and that for local time is 24 hours. For subsequent months and years, the initial analysis serves as a sliding data window. To find subsequent monthly values, this window is advanced by one month, and the algorithm is applied again. Further details can be found in Huang et al.,[2010a]."

3): Referee #1: The analysis provides the characteristics of the tides and their evolution over time. There is a detailed discussion of comparisons of temperature trends inferred from SABER data with published results from terrestrial lidars and AMSU satellites for several fixed local hours. These comparisons are very interesting and show a good general agreement, but results on the decadal evolution of tidal parameters (amplitude and phase) are not presented, except for the example given in Figure 2. I do not know any published results on the evolution of tidal parameters in the middle atmosphere and this would be a very valuable result.

Authors' response to 3): We have added an Appendix to the manuscript, which contain additional plots of tidal trends, corresponding to Figure 2b of the manuscript. The Appendix is included below here also. The narrative of the Appendix is as follows: "We present additional figures, corresponding to Figure 2 (b), of temperature diurnal amplitudes and phases over more altitudes (20, 40, 60, 80, 90 km) and latitudes (0⁰, 40⁰). The left panels (a) of each figure show temperature tidal diurnal amplitudes and phases at various altitudes and the Equator, while the right panels (b) correspond to the left panels but at 40⁰N latitude. In each panel, the left axis scale and black line denote tidal diurnal amplitudes (K), while the right axis scale and red line show the diurnal phases (hr of maximum value). The displayed trend values are obtained from a simple least squares straight line fit. The larger variations generally reflect modulation of the tides by the quasi biennial oscillation (QBO). This has been discussed in models (Mayr and Mengel, 2005) and other SABER data (Forbes et al., 2008). Although the figures may give additional insight to the nature of the trends, there are caveats to be considered. We note that the semidiurnal amplitudes and phases can also be significant. We have derived a total of five Fourier components, and our numerical results reflect all 5 Fourier terms. Because both amplitudes and phases exhibit trends, they need to be considered in parallel, in tandem, and this is difficult to discern, qualitatively. In addition, because the trends are generally small, it would be difficult to arrive at conclusions."

4): Referee #1: Concerning the organisation of the paper, it is rather unusual to give some conclusions in the introduction section as it is done in lines 81-89. This part should be moved to the summary and conclusion section.

4) Authors response: We have removed the paragraph.
* * *
5) Appendix

Please see supplement file for figures.

We present additional figures, corresponding to Figure 2 (b), of temperature diurnal amplitudes and phases over more altitudes (20, 40, 60, 80, 90 km) and latitudes (0⁰, 40⁰). The left panels (a) of each Figure show temperature tidal diurnal amplitudes and phases at different altitudes and the Equator, while the right panels (b) correspond to the left panels but at 40⁰N latitude. In each panel, the left axis scale and black line denote tidal diurnal amplitudes (K), while the right axis scale and red line show the diurnal phases (hr of maximum value). The displayed trend values are obtained from a simple least squares straight line fit. The larger variations generally reflect modulation of the tides by the quasi biennial oscillation (QBO). This has been discussed in models (Mayr and Mengel, 2005) and other SABER data (Forbes et al., 2008). Although the figures may give additional insight to the nature of the trends, there are caveats to be considered. We note that the semidiurnal amplitudes and phases can also be significant. We have derived a total of five Fourier components, and our numerical results reflect all 5 Fourier terms. Because both amplitudes and phases exhibit trends, they need to be considered in parallel, in tandem, and this is difficult to discern, qualitatively. In addition, because the trends are generally small, it would be difficult to arrive at conclusions.

Figure A1. Left panel (a): Temperature tidal diurnal amplitudes and phases at 20 km, equator; left axis scale: black line: tidal diurnal amplitude (K); right axis scale: red line: diurnal phase (hr of maximum value). Right panel (b): as in left panel but at 40⁰N
latitude.

Figure A2. Left panel (a): Temperature tidal diurnal amplitudes and phases at 40 km, equator; left axis scale: black line: tidal diurnal amplitude (K); right axis scale: red line: diurnal phase (hr of maximum value). Right panel (b): as in left panel but at 40⁰N latitude.

Figure A3. Left panel (a): Temperature tidal diurnal amplitudes and phases at 60 km, equator; left axis scale: black line: tidal diurnal amplitude (K); right axis scale: red line: diurnal phase (hr of maximum value). Right panel (b): as in left panel but at 40⁰N latitude.

Figure A4. Left panel (a): Temperature tidal diurnal amplitudes and phases at 80 km, equator; left axis scale: black line: tidal diurnal amplitude (K); right axis scale: red line: diurnal phase (hr of maximum value). Right panel (b): as in left panel but at 40⁰N latitude.

Figure A5. Left panel (a): Temperature tidal diurnal amplitudes and phases at 90 km, equator; left axis scale: black line: tidal diurnal amplitude (K); right axis scale: red line: diurnal phase (hr of maximum value). Right panel (b): as in left panel but at 40⁰N latitude.

Please also note the supplement to this comment:
https://angeo.copernicus.org/preprints/angeo-2020-63/angeo-2020-63-AC1-supplement.pdf

---

## Author Comment (AC2) · 14 Dec 2020

December 14, 2020

Response to anonymous reviewer #2 interactive comment concerning manuscript titled "Temperature decadal trends, and their relation to diurnal variations in the lower thermosphere, stratosphere, and mesosphere, based on measurements from SABER on TIMED" by Frank T. Huang and Hans G. Mayr

Anonymous Referee #2 A) Referee #2: The authors have used temperatures obtained from SABER data to extend study about the local time variations of temperature trends from 2002 to 2014, 20 to 100 km, and 48_S to 48_N latitude. The trends found have been compared with those published and after some discussion the authors found that temperature decadal trends for a fixed local time were different from trends at another fixed local time. In addition, the authors also found that the amplitudes and phases of the tides also revealed decadal trends, and they inferred that thermal tides likely count to contribute to the local time variations of temperature trends.

Based on my evaluation, this is an important scientific contribution which can help to clarify differences and achieve more consistent trend results. However, there are some concerns that need to be addressed.

1) Referee #2: comment 1): The introduction can be improved in order to clarify the findings in previous works and the difference from present investigation. The authors have discussed this issue, however the way it is written has become confused.

Authors' response to comment 1: We are also a bit confused. In the introduction, there is one reference which we have moved. Otherwise, we do not mention previous works. We have now added sentences that refer to our previous works, and to Section 2, which contains the details of our descriptions of previous work.

2) Referee #2: comment 2): One of the main results found by the authors concerns the contribution of thermal tides to the local time variations of temperature trends, since their amplitudes and phases also display decadal trends. In this sense, Figure 2b should be further explored in order to show readers how the variation in thermal tides contributes to the local time variations of temperature trends. The same can be considered in sections 4.1 (Stratosphere) and 4.2 (Lower Thermosphere), where the relationship between the thermal tides and the trends should be emphasized.

Authors' response to comment 2: We have added an Appendix to the manuscript,
which contain additional plots of tidal trends, corresponding to Figure 2b of the manuscript. The Appendix is included below here also. The narrative of the Appendix is as follows: "We present additional figures, corresponding to Figure 2 (b), of temperature diurnal amplitudes and phases over more altitudes (20, 40, 60, 80, 90 km) and latitudes (0âĄř, 40âĄř). The left panels (a) of each Figure show temperature tidal diurnal amplitudes and phases at different altitudes and the Equator, while the right panels (b) correspond to the left panels but at 40âĄřN latitude. In each panel, the left axis scale and black line denote tidal diurnal amplitudes (K), while the right axis scale and red line show the diurnal phases (hr of maximum value). The displayed trend values are obtained from a simple least squares straight line fit. The larger variations generally reflect modulation of the tides by the quasi biennial oscillation (QBO). This has been discussed in models (Mayr and Mengel, 2005) and other SABER data (Forbes et al., 2008). Although the figures may give additional insight to the nature of the trends, there are caveats to be considered. We note that the semidiurnal amplitudes and phases can also be significant. We have derived a total of five Fourier components, and our numerical results reflect all 5 Fourier terms. Because both amplitudes and phases exhibit trends, they need to be considered in parallel, in tandem, and this is difficult to discern, qualitatively. In addition, because the trends are generally small, it would be difficult to arrive at conclusions."

3) Referee #2: comment 3): The diurnal tide on temperature in stratosphere (_ 40-50 km) has its strongest amplitudes at latitudes 40_N, S, while in the MLT region (_ 85-95 km) they are strongest around the equator, where the phases are best resolved. Therefore, I would like to suggest adding the analysis of trends in the amplitudes and phases of the diurnal tide for these regions, which may provide enrichment in the discussion of results.

Authors' response to comment 3: The plots at 90 km and the Equator, and at 40 km at 40deg, are shown below in the Appendix, as described in the response to C) earlier. Although the plots can provide valuable insight, we believe that numerical analyses are

needed, and some sort of modeling. Currently we are not aware of 3D models focusing on trends related to local times. If there are, a simple test would be to constrain zero trends on the tides, and see if there are effects on the trends of the temperature.

4) Referee #2: comment 4) The theme of the section 4.3 (Orbital drift and generic) needs to be improved and explored. It could be moved to section 2 without prejudice to the manuscript.

Authors' response to comment 4: We realize that Sections 4.3 and 4.4 are not only brief, but even skimpy. For the general dependence of trends on local time, because the results are new, our motive was only to provide more information on a wider altitude and latitude range. Because trends depend on the time span considered, the tidal trends may well also be different for another time span, and their behavior relative the mean temperature trend may also be different. As noted in replies to comments 2 and 3, we believe that further studies would need to entail numerical aspects, and some sort of modeling. Again, if there are 3D models which focus on trends and local times, a simple test would be to constrain zero trends on the tides, and see if there are effects on the trends of the temperature For orbital drifts, there have been no previous similar results on this subject. Our motive is only meant to provide an indication of what may result when local times at which measurements are made are not controlled. Specifics would depend on the drift of the particular satellite and the particular study. Due also to other comments, we have merged Sections 4.3 and 4.4, and added a bit more information.

5) Referee #2: comment 5: Discussion should be made more rigorous. The basis for the statements needs to expand further.

Authors' response: We have added to the discussion, describing the basis of our calculation

The discussion is now as follows: "Using SABER data, we have investigated the local time variations of temperature trends (K/decade) from 2002 to 2014, 20 to 100 km, and

48°S to 48°N latitude. SABER provides global temperature measurements over the 24 hrs of local time, and from 20 to 100 km in altitude, that are not available from other satellites and sources. From our past studies based on SABER data, we had estimated diurnal variations of the temperature (thermal tides) for each day, expressed in the form of five Fourier series components (Huang et al., 2010a). We had also derived zonal means of temperature that are averages over both longitude and local time for a latitude circle (Huang et al., 2006). These 'synoptic' zonal means are important because it can then be compared directly with 3D models (Austin et. al., 2008). As explained earlier, zonal means from sun-synchronous satellites are tied to one or two local times. To our knowledge, comparable zonal means of temperature that are averages over longitude and the 24 hours of local time are just not available elsewhere. In this current study, we have combined the past results to estimate the zonal mean trends corresponding to specific local times. These results at local times have not been available previously. They show that the values of temperature decadal trends for a fixed local time are different from trends at another fixed local time. We find that the amplitudes and phases of the tides themselves also display decadal trends and are then likely contributors to the local time variations of temperature trends. Our results of trend variations with local time are supported by comparisons with corresponding nighttime lidar measurements in the stratosphere and lower thermosphere. They are also supported by comparisons with corresponding satellite measurements made at specific local times in the stratosphere. The dependence of trends on local time is significant throughout the region of analysis, and can be significant even from hour to hour, as can be seen in Figures 3, 4, 5, and 6. In the lower thermosphere, this agrees with corresponding trend results by She et al.,[2019], based on lidar night-time measurements. She et al., [2019] found that trends based on a two-hour average near midnight show systematic differences from the average over other hours. Our comparisons with the overnight results of She et al., [2019] are seen in Figure 5, where our trends at 19, 20, and 21 hours compare favorably, while our day time trends at 15, 16, and 17 hours compare less favorably. In the stratosphere, our comparison with trends found by Funatsu et al.,[2016], based

on lidar and AMSU measurements, are even better, as seen in Figures 3 and 4. At 44°N (AMSU and OHP lidar), Funatsu et al., [2016] provide AMSU trend results only from 30 to 40 km, but they match our results almost exactly. Their results from 20 to 40 km, representing mid latitudes (30° to 60°N) also match our results almost exactly from 20 to 30 km, but are larger from 30 to 40km. Between ∼ 30 to 40 km, the night-time lidar trends are significantly smaller (more negative) than both our and that of Funatsu et al.,[2016]. However, when the comparison is between night time lidar and our night-time results (21, 22, and 23 hours, see Figures 3a, 3b), the agreements are better. At 20°N (AMSU and MLO lidar), similar comments apply. These examples all suggest that at least some of the differences between night time lidar trends and those based on other measurements that are not made at night, can be explained at least partly, through variations of trends with local time. However, we emphasize that our three examples of course do not a pattern make, and more direct comparisons are needed. Our current comparisons are limited because the various results should be based on the similar time spans, and also not based on merged data from various sources, as the identity in local time would not be clear for merged data. Although there have been previous studies related to variations with local time, they focused on mitigating differences when merging data from different sources, and on accounting for temperature variations with local time due to orbital drifts. Because our results show that the data sets representing measurements at different fixed local times can result in varying trends, merging those data can result in trends that cannot be tied to specific local times, or to averages over the 24 hours of local time, as in 3D models, and can result in biases."

Referee #2: Technical revision Please check the citations through the text considering the rules required by the journal. Authors: Will do.

Referee #2: Page 2, line 82: change "Advanced Microwave Sounder..." by "Advanced Microwave Sounder Unit" Authors: The paragraph containing the sentence has been removed because the other referee believed that conclusions such as that expressed

there do not belong in the introduction.

Referee #2: Page 7, Figure 3b,c,d: legend overlap with plots Page 11, Figure 7: legend overlap with plots

Authors: We have re-plotted the figures.
* * *
______-

Appendix

Please see supplement file for figures.

We present additional figures, corresponding to Figure 2 (b), of temperature diurnal amplitudes and phases over more altitudes (20, 40, 60, 80, 90 km) and latitudes (0âĄř, 40âĄř). The left panels (a) of each figure show temperature tidal diurnal amplitudes and phases at various altitudes and the Equator, while the right panels (b) correspond to the left panels but at 40âĄřN latitude. In each panel, the left axis scale and black line denote tidal diurnal amplitudes (K), while the right axis scale and red line show the diurnal phases (hr of maximum value). The displayed trend values are obtained from a simple least squares straight line fit. The larger variations generally reflect modulation of the tides by the quasi biennial oscillation (QBO). This has been discussed in models (Mayr and Mengel, 2005) and other SABER data (Forbes et al., 2008). Although the figures may give additional insight to the nature of the trends, there are caveats to be considered. We note that the semidiurnal amplitudes and phases can also be significant. We have derived a total of five Fourier components, and our numerical results reflect all 5 Fourier terms. Because both amplitudes and phases exhibit trends, they need to be considered in parallel, in tandem, and this is difficult to discern, qualitatively. In addition, because the trends are generally small, it would be difficult to arrive at conclusions.

Figure A1. Left panel (a): Temperature tidal diurnal amplitudes and phases at 20 km,

equator; left axis scale: black line: tidal diurnal amplitude (K); right axis scale: red line: diurnal phase (hr of maximum value). Right panel (b): as in left panel but at 40âĄřN latitude.

Figure A2. Left panel (a): Temperature tidal diurnal amplitudes and phases at 40 km, equator; left axis scale: black line: tidal diurnal amplitude (K); right axis scale: red line: diurnal phase (hr of maximum value). Right panel (b): as in left panel but at 40âĄřN latitude.

Figure A3. Left panel (a): Temperature tidal diurnal amplitudes and phases at 60 km, equator; left axis scale: black line: tidal diurnal amplitude (K); right axis scale: red line: diurnal phase (hr of maximum value). Right panel (b): as in left panel but at 40âĄřN latitude.

Figure A4. Left panel (a): Temperature tidal diurnal amplitudes and phases at 80 km, equator; left axis scale: black line: tidal diurnal amplitude (K); right axis scale: red line: diurnal phase (hr of maximum value). Right panel (b): as in left panel but at 40âĄřN latitude.

Figure A5. Left panel (a): Temperature tidal diurnal amplitudes and phases at 90 km, equator; left axis scale: black line: tidal diurnal amplitude (K); right axis scale: red line: diurnal phase (hr of maximum value). Right panel (b): as in left panel but at 40âĄřN latitude.

Please also note the supplement to this comment:
https://angeo.copernicus.org/preprints/angeo-2020-63/angeo-2020-63-AC2-supplement.pdf